# Characterization of the Intracellular Acidity Regulation of Brain Tumor Cells and Consequences for Therapeutic Optimization of Temozolomide

**DOI:** 10.3390/biology12091221

**Published:** 2023-09-08

**Authors:** Alaa Tafech, Pierre Jacquet, Céline Beaujean, Arnold Fertin, Yves Usson, Angélique Stéphanou

**Affiliations:** Univ. Grenoble Alpes, CNRS, UMR 5525, VetAgro Sup, Grenoble INP, TIMC, 38000 Grenoble, France; alaa.a.tafech@gmail.com (A.T.); pierre.jacquet1@univ-grenoble-alpes.fr (P.J.); celine.beaujean@univ-grenoble-alpes.fr (C.B.); arnold.fertin@univ-grenoble-alpes.fr (A.F.); yves.usson@univ-grenoble-alpes.fr (Y.U.)

**Keywords:** BCECF probe, fluorescence microscopy, glioblastoma, pH regulation, resistance to acidity, temozolomide

## Abstract

**Simple Summary:**

Tumor cells have high levels of metabolic activity, which makes the environment around them more acidic. This acidic environment encourages the aggressiveness and invasiveness of the tumor, which are linked to a worse prognosis. Cancer cells may control their internal acidity in different ways when exposed to an acidic environment. In this study, two types of brain tumor cells were exposed to different levels of acidity to characterize their response in terms on intracellular pH regulation. The results show that these cells do not handle acidity the same way. One cell type can reduce its internal acidity, while the other can withstand higher acidity levels. The ability to manage acidity seems to depend on the specific cell type and involves various mechanisms to keep the cells working properly. This study explored these cells in both 2D flat layers and 3D structures called spheroids. In spheroids, the regulation process of internal acidity appears to work differently than in 2D. These findings could influence treatment because the drug temozolomide, which is used to fight brain tumors, works better in certain pH environments. By adjusting the pH outside the cells, the effectiveness of the drug can be improved. This suggests that personalized treatment might be possible by combining temozolomide with substances that control pH.

**Abstract:**

A well-known feature of tumor cells is high glycolytic activity, leading to acidification of the tumor microenvironment through extensive lactate production. This acidosis promotes processes such as metastasis, aggressiveness, and invasiveness, which have been associated with a worse clinical prognosis. Moreover, the function and expression of transporters involved in regulation of intracellular pH might be altered. In this study, the capacity of tumor cells to regulate their intracellular pH when exposed to a range of pH from very acidic to basic was characterized in two glioma cell lines (F98 and U87) using a new recently published method of fluorescence imaging. Our results show that the regulation of acidity in tumors is not the same for the two investigated cell lines; U87 cells are able to reduce their intracellular acidity, whereas F98 cells do not exhibit this property. On the other hand, F98 cells show a higher level of resistance to acidity than U87 cells. Intracellular regulation of acidity appears to be highly cell-dependent, with different mechanisms activated to preserve cell integrity and function. This characterization was performed on 2D monolayer cultures and 3D spheroids. Spatial heterogeneities were exhibited in 3D, suggesting a spatially modulated regulation in this context. Based on the corpus of knowledge available in the literature, we propose plausible mechanisms to interpret our results, together with some new lines of investigation to validate our hypotheses. Our results might have implications on therapy, since the activity of temozolomide is highly pH-dependent. We show that the drug efficiency can be enhanced, depending on the cell type, by manipulating the extracellular pH. Therefore, personalized treatment involving a combination of temozolomide and pH-regulating agents can be considered.

## 1. Introduction

Acidity is a well-known feature of the tumor microenvironment. It is observed in a variety of solid tumors [1,2,3]. pH levels can be very heterogeneous within the same tumor, with localized acid zones. pH measurements performed using electrodes have shown that pH can reach 5.9 in brain tumors, with an average value of around 6.8, while the pH in normal brain tissue is around 7.1 [1].

Tumor cells that proliferate abnormally quickly run out of oxygen and become hypoxic. In hypoxia, the cells are not able to produce ATP via OXPHOS. In order to adapt to this harmful environment, HIF signaling shifts energy production towards anaerobic glycolysis, allowing hypoxic tumor cells to continue to produce ATP despite the low oxygen levels. Due to their significant fermentation activity, hypoxic tumor cells secrete a large amount of lactate and H${}^{+}$ ions, which can lead to changes in the intracellular pH (pH${}_{i}$) of cancer cells. To help deal with the excess production of lactate and H${}^{+}$ ions, tumor cells activate a number of pH-regulating proteins to export lactate and H${}^{+}$ ions and keep the pH${}_{i}$ within the physiological level. Once excreted, lactate and H${}^{+}$ ions induce a decrease in extracellular pH (pH${}_{e}$), leading to acidification of the tumor microenvironment [3,4,5,6].

Tumor cells are generally associated with alkaline pH${}_{i}$ values of 7.1 to 7.6 and acidic pH${}_{e}$ values of 6.2 to 6.9 [7], whereas normal cell have lower pH${}_{i}$ levels (7.0–7.2) than those found in the environment under physiological conditions (7.3–7.4) [8,9]. As a consequence, there is an inverse pH gradient for tumor cells. The fact that pH${}_{i}$ is significantly higher than pH${}_{e}$ in tumors demonstrates the existence of powerful mechanisms that prevent acidification of the intracellular environment [10,11,12]. The tools used by tumor cells to regulate their pH${}_{i}$ are varied and depend on the cell type. Several pH regulatory proteins show increased expression (often modulated by transcription factor HIF-1) or activity in tumor cells. These redundant proteins of tumor pH regulation include Na${}^{+}$/H${}^{+}$ exchangers (NHEs) such as NHE1 [13]; vacuolar-type adenosine triphosphatase (V-ATPase) [14]; monocarboxylate transporters (MCTs) such as MCT1, MCT2, MCT3, and MCT4 [15]; carbonic anhydrases such as CAII, CAIX, and CAXII [13,16]; and the HCO${}_{3}^{-}$ transporters [17]. Distinct mechanisms can also be set up in cells with different invasive or metastatic potentials [18,19]. A different subcellular localization of pH regulators was demonstrated in a model of brain tumors [20].

It has become evident that pH${}_{i}$ is an important regulator of metabolism and many cell functions, such as cell proliferation [21,22,23], as well as cell cycle progression [24,25] and differentiation [26,27]. The inverted pH gradient is considered a hallmark of cancer [28], which promotes tumor growth, invasion, and metastasis of cancer cells via various mechanisms. Therefore, it appears that the measurement of pH${}_{i}$ in tumors can be of interest in monitoring the progression of cancers and the responses of cancer cells to various treatments.

Despite advances in the development of new therapeutic strategies, a major underlying factor in cancer-related death remains treatment resistance. The cytoplasmic membrane functions as a semipermeable physical barrier that separates the intracellular and extracellular environments. Small and uncharged molecules can easily diffuse through the membrane, while the passage of charged molecules is more difficult, since the membrane is polarized. On the other hand, the diffusion of chemotherapeutic agents can be altered by changes in the pH of the tumor environment. For instance, an acidic pH${}_{e}$ increases the cellular uptake of weakly acidic drugs such as chlorambucil [29] and melphalan [30], thereby increasing their efficacy. In contrast, acidic pH${}_{e}$ retards the uptake of weakly basic drugs such as doxorubicin [31,32] and mitoxantrone [33,34], and the extracellular acidity that can be found in the tumor microenvironment can limit their passage through the membrane of cancer cells, compromising their antitumor efficacy. Extracellular acidity may also play a role in drug resistance by increasing the activity of P-glycoprotein (P-gp), which is involved in drug efflux. Activation of the p38 MAP kinase signaling pathway by acidity is involved in the activation of P-glycoprotein. Therefore, inhibition of the p38 pathway restores the sensitivity of tumor cells to chemotherapeutic agents.

Cancer drug research and development initially focused on targeting cancer cell proliferation. It then appeared that the environment could also be taken into account. The role of acidity in the evolution and therapeutic response of tumors has been particularly documented, and considerable attention is currently focused on strategies targeting this tumor acidity. Different strategies that directly target tumor acidity or exploit characteristics of acidity are now being developed and have begun to show promising results.

A first strategy targeting tumor acidity consists of studying drugs that can be activated under acidic conditions to improve the specificity of the treatment. For example, the molecule temozolomide (TMZ) has been used against glioblastoma [35]. Another strategy is to inhibit the various proteins involved in maintaining pH${}_{i}$, which is crucial for the survival of tumors in an acidic environment, using small molecule inhibitors or antibodies. Targeted proteins include NHEs, MCTs, CAs, NBCs (Na${}^{+}$/HCO${}^{-}$ cotransporters), and V-ATPase [36,37,38]. Tumor acidity is also exploited in the development of nanoparticles, which release cytotoxic agents under acidic conditions encountered in tumor tissue [39,40].

In this study, we characterized the regulation of intracellular acidity of brain tumor cells for two cell lines: F98 murine glioma and U87 human glioblastoma. Characterizations were performed using fluorescence microscopy with the a BCECF pH-dependent fluorescent probe, for which we recently developed a new methodology that extends the accuracy and range of pH measurements [41]. Measurements were performed on 2D cell cultures and on 3D spheroids. Differences were highlighted between the two cell lines, and some hypotheses were proposed to explain them. Finally, the impact of the extracellular pH on TMZ efficacy was investigated in the two cell lines with the aim of identifying the optimum pH range for each.

## 2. Materials and Methods

### 2.1. Cell Lines

U-87 MG and F98 cell lines from ATCC were used. The U87 cell line was established from a human glioblastoma. The cells are often polynuclear with very fine membrane extensions. Their mean doubling time is 30.8 ± 2.5 [42]. The F98 cell line is a murine glioma. The cells are predominantly spindle-shaped and have an in vitro doubling time of approximately 18 h [43]. Both cell lines were cultured in the same conditions in plastic flasks. The culture medium was composed of DMEM with 4.5% glucose supplemented with 10% fetal bovine serum and 2 mM glutamine (Dutscher, Bernolsheim, France). The cells were maintained at 37 °C in a humid atmosphere with 5% CO${}_{2}$. For the experiments, cells were seeded at a concentration of 1000 cells/well in 96-well flat CleaLine^®^ plates for 2D monolayer cultures or in 96-well CELLSTAR^®^ round-bottom ultra-low attachment plates for 3D spheroid cultures.

### 2.2. Microscope

A Zeiss LSM 710 laser scanning confocal microscope was used. It is an inverted microscope especially suited for live cell imaging experiments equipped with an incubation chamber to maintain physiological conditions at a temperature of 37 °C under 5% CO${}_{2}$. It is equipped with the following 6 laser lines: diode, 405 nm; argon, 458 nm, 488 nm, and 514 nm; and helium–neon, 543 nm and 633 nm.

### 2.3. pH Measurements

A BCECF fluorescent pH-sensitive probe (2′,7′-bis(carboxymethyl)-5(6)-carboxyfluorescein) purchased from Thermo Fischer Scientific, Bourgoin-Jallieu, France was used for pH measurements. The fluorescent excitation profile is pH-dependent. The shape of the emission spectra remains almost constant with a main emission peak. A ratiometric pH measurement was implemented so that pH determination was independent of the probe concentration or optical path length because pH is related to the ratio (in excitation or emission) at two characteristic wavelengths. In our case, two excitation wavelengths were used, and the fluorescence emission was measured at one wavelength for each excitation. The ratio (*R*) of the fluorescent intensities measured for these two emission wavelengths was monitored as a function of the pH to construct the R=f(pH) curve (see Appendix A). For our measurements, we exploited the best combination of emission wavelengths obtained with dual excitation at 405 nm and 488 nm based on a methodology we recently developed to optimize the quality of measurements [41]. We have found that the best method to measure the intracellular pH of F98 cells is to set the emission wavelength to 517 nm for the 405 nm laser and 546 nm for the 488 nm laser. In this case, the pH is monitored as a function of the fluorescence ratio (R = I${}_{517\mathrm{nm}}$/I${}_{546\mathrm{nm}}$), whereas for U87 cells, the intracellular pH is measured by fixing the emission wavelength at 556 nm for the 405 nm laser and at 517 nm for the 488 nm laser. In this case the pH is monitored as a function of to the fluorescence ratio (R = I${}_{517\mathrm{nm}}$/I${}_{556\mathrm{nm}}$).

We note that for intracellular pH measurements, BCECF-AM was used. The added acetoxymethyl ester (AM) group makes the probe neutral, allowing it to cross cell membranes. Once inside the cell, intracellular esterases cleave the ester group of the molecule, releasing the charged groups and trapping the probe in the cytoplasmic compartment [44,45]. The BCECF devoid of an AM group cannot enter the cells and remains in the extracellular compartment.

### 2.4. Loading Cells with pH Probes

The fluorescent probe was loaded into cells thanks to its cell-permeable ester derivative that is hydrolyzed by cytosolic esterases to yield fluorescent intracellularly trapped probes. Suspended cells were cultured in 96-well, flat-bottom, ultra-low attachment plates (Dominique Dutscher, Bernolsheim, France). They were seeded on day 1 at a density of 1 × 10${}^{3}$ cells per well in a volume of 200 μL of complete medium and allowed to adhere for 48 h before the experiment. Then, cells were incubated in serum-free DMEM with 5 μM of probe prepared from a 1 mM stock solution in DMSO for 30 min at 37 °C under 5% CO${}_{2}$. FBS serum may exhibit endogenous esterase activity. Therefore, loading medium should be serum-free to keep extracellular hydrolysis of the AM ester to a minimum. Once loaded with the probe, cells were washed three times at 37 °C to remove the probe not taken up by the cells using 200 μL of serum-free DMEM at the pH desired for the observation.

### 2.5. Intracellular pH Calibration

Calibration of the intracellular pH after loading the cells with the pH probe is based on an equilibrium of the intracellular pH (pH${}_{i}$) with the extracellular pH (pH${}_{e}$) of the culture medium, the value of which is known. In other words, it consists of imposing the pH${}_{i}$ at known values and determining the fluorescence ratio for each of these values (as described above). To impose the pH${}_{i}$, the addition of nigericin in the culture medium, the ionic composition of which in K${}^{+}$ is similar to that of the treated cells, allows for equilibration of the intracellular pH with the pH of the medium by creating a membrane antiport K${}^{+}$/H${}^{+}$. Therefore, as a first step, the calibration solutions are prepared so that they contain a concentration of 90 mM of K${}^{+}$ ions and 5 μM of nigericin. The pH is adjusted by adding 0.1 M HCl or 0.1 M NaOH in order to obtain solutions in a pH range of 4 to 9. For each cell line, the ratio of fluorescence intensities is monitored as a function of pH, and the $R=f({\mathrm{pH}}_{i})$ curve is constructed (see Appendix A).

### 2.6. pH Measurements in Spheroids

Spheroids were cultured in CELLSTAR^®^ 96-well, round-bottom, ultra-low attachment plates. F98 and U87 cells were seeded on day 1 at a density of 1 × 10${}^{3}$ cells per well in a volume of 200 μL of medium. The cells were maintained under standard culture conditions in an incubator at 37 °C in a humid atmosphere composed of 95% air and 5% CO${}_{2}$. Spheroids were formed 24 h after plating (1 spheroid per well).

pH${}_{i}$ measurements were initiated on day 4 by adding 5μM of BCECF-AM to the culture medium. After 60 min of incubation with BCECF-AM, spheroids were washed 3 times with fresh medium (pH∼7.4) to remove the probes not taken up by the cells. The spheroids were then exposed to different DMEM media with three different pH${}_{e}$ values (5.00, 6.00, and 7.40).

Eight hours after incubation with a given pH${}_{e}$, confocal fluorescence images were recorded by successively exciting each spheroid at 405 nm and 488 nm. Starting from one pole of each spheroid, the excitations were carried out in optical sections along the Z axis (confocal z stack) through the spheroid with a Z-slice spacing of 5 μm between adjacent optical planes. With our confocal setup, light penetration into the spheroid was limited to a 150 μm depth.

To monitor the pH${}_{i}$ in spheroids, the fluorescence images were processed as follows. The background signal taken from an empty region was subtracted from the measurements, and the ratio of emission intensity resulting from excitation at the two wavelengths (405 nm and 488 nm) was calculated on a pixel-to-pixel basis. To convert the ratio signals to pH${}_{i}$ values in the spheroids, the appropriate calibration curves obtained for 2D cell cultures were used.

We note that pH${}_{e}$ values could not be measured in the interstitial liquid between the cells inside the spheroids, since we could not detect any extracellular BCECF signal in the spheroid mass. This may be related to the limited diffusion of the BCECF molecules through the spheroid mass into the interstitial liquid due to the high cell density.

In total, 18 spheroids were treated, including the F98 and U87 cell lines. All measurements were performed based on three replicates for each pH${}_{e}$ value considered (5.00, 6.00, and 7.40) and for each cell line. The most representative replicate in each case is presented for illustration.

### 2.7. Assessment of Cell Resistance to Acidity

In order to assess the cells’ in vitro resistance to pH${}_{e}$ changes, cells were seeded at a concentration of 5 × 10${}^{4}$ cells/mL in a 24-well plate and incubated for 48 h at 37 °C in a humid atmosphere containing 5% CO${}_{2}$. After 48 h of growth, the culture medium was removed and replaced with a test medium with a pH${}_{e}$ ranging from 4 to 9. To assess cell mortality, the trypan blue stain was used. Trypan blue stains dead and dying cells whose membrane integrity is damaged. The culture medium was removed from each well, and 200 μL of trypsin was added to each well to remove the cells. The removed cells were then resuspended in 800 μL of culture medium. The cell suspensions were diluted 1:1 (*v*:*v*) with 0.4% trypan blue; then, a 10 μL mixture was transferred to a cell count slide. Cell mortality was monitored using an automated cell counter over a period of 72 h, and the number of dead cells was measured every four hours for each tested pH${}_{e}$ value.

### 2.8. Measure of Cell Viability after TMZ Treatment

Viability after temozolomide (TMZ) treatment under different ${\mathrm{pH}}_{e}$ values was evaluated using an MTT test. The cells were seeded on day 1 in a 96-well plate at a rate of 1 × 10${}^{4}$ cells per well in 200 μL of culture medium. On day 3, the cells were incubated for 48 h with TMZ with different extracellular pH values for a final volume of 100 μL per well. At the end of the incubation period, 10 μL of an MTT solution was added to achieve a final concentration of 0.5 mg/mL. The cells were then incubated for 3 h at 37 °C under 5% CO${}_{2}$ and protected from light. The formazan crystals that formed after incubation were dissolved by adding 100 μL of dimethyl sulfoxide to each well. Finally, the amount of live cells was quantified by an absorbance reading at 570 nm. The viability is expressed as a percentage of living cells relative to the control.

In spheroid cultures, cell viability was evaluated by confocal microscopy using two probes: the BCECF probe, which was used as a powerful probe that allows for labeling of all the cells, and the Sulforhodamine B probe (SRB, purchased from Thermo Fischer Scientific), which marks dead cells in red. The difference between these two fluorescent signals provides the relative amount of living cells in the spheroid. The spheroids were grown for 3 days, then exposed to TMZ for 48 h. The spheroids were then incubated for one hour with 5 μM of BCECF and 10 μM of SRB before image acquisition.

### 2.9. Evaluation of IC${}_{50}$ of TMZ

The TMZ IC${}_{50}$ value, i.e., the TMZ concentration that results in 50% cell viability, was evaluated for the two cell lines using an MTT test. The cells were incubated in standard culture medium at physiological pH = 7.4 for 48 h with the following TMZ concentrations: 0, 50, 100, 200, 300, 400, 500, and 1000 μM. Viability was measured as the mean of 3 replicates for each tested concentration. The IC${}_{50}$ values of 200 μm and 250 μm were measured for F98 and U87 cells, respectively.

### 2.10. TMZ Treatment

The cell lines in 2D cell culture and in spheroids were exposed to TMZ for 48 h. TMZ was solubilized using DMSO. The experiments were carried out under the three following conditions: with TMZ, with DMSO only, and with neither TMZ or DMSO. DMSO was considered the control in order to evaluate the effect of TMZ only.

### 2.11. Statistical Analysis

All experiments were performed with three replicates. All the data points are presented as the mean and standard deviation (SD) of these three replicates. The number of replicates was not sufficient to perform a reliable statistical analysis. As a consequence, the presented results should be considered an overall trend, not a definitive confirmation. We made one exception for the evaluation of temozolomide, since we performed pairwise experiments amenable to a statistical paired *t*-test that we employed to compare the mean of 3 paired biological repeats. Significant differences are defined as *p* < 0.05.

## 3. Results

### 3.1. Intracellular pH Changes in Cell Cultures

The F98 and U87 cell line were first exposed to instantaneous extracellular pH changes. At time t=0 the extracellular pH (pH${}_{e}$) of the culture medium with a pH of 7.4 was changed by a culture medium with a new pH${}_{e}$. Eighteen different pH${}_{e}$ values were considered from 5 to 8.4 in 0.2 increments. For each new pH${}_{e}$, the changes in the intracellular pH (pH${}_{i}$) were monitored every hour over a period of 8 h. Confocal fluorescence images were recorded by successively exciting the BCECF probe internalized by the cells at 405 nm and 488 nm.The intensity ratio of the two emission wavelengths was calculated on a pixel-to-pixel basis, and the pH${}_{i}$ was determined using the calibration curves (Appendix A).

Parallel measurements were conducted to measure the evolution of the pH${}_{e}$ using the non-permeable version of the BCECF. It is known that the pH of the culture medium is not stable over time unless a buffer is used [41]. Since we aimed to characterize spontaneous pH${}_{i}$ adaptation, we chose not to regulate the pH${}_{e}$ with a buffer in order to avoid constraining the overall system.

Beforehand, the basal pH${}_{i}$ of each cell line was measured in its physiological culture medium at pH${}_{e}$ = 7.4. Under this initial basal condition, pH${}_{i}$ was 7.65 ± 0.13 for F98 cells and 7.48 ± 0.12 for U87 cells.

Figure 1A,B show the effect of pH${}_{e}$ change on the pH${}_{i}$ of F98 and U87 cells as a function of time. In the short term (∼1 h), the sudden change in pH${}_{e}$ creates a shock for the cells, since the pH${}_{i}$ of both cell lines almost instantly takes the value of the pH${}_{e}$. Then, the pH${}_{i}$ regulation process takes 2 h to produce an effect and one more hour to reach a stationary state. In the longer term, after 4 h and for up to 8 h, the pH${}_{i}$ remains stable and is specific to the cell line.

Figure 1C,D show the simultaneous measurements of pH${}_{e}$, which exhibit a similar drift for both cell culture media. At the time scale of 8 h, the variations in the pH${}_{e}$ are independent of the presence of the cells [41].

By combining Figure 1A,C and Figure 1B,D, a pH${}_{e}$ value can be associated with a corresponding pH${}_{i}$ value measured at the same instant. This relationship between pH${}_{e}$ and pH${}_{i}$ is displayed in Figure 2 for the two cell lines, allowing the pH${}_{e}$ drift to be integrated.

#### 3.1.1. pH Changes in F98 Cells

For F98 cells, the pH${}_{i}$ is almost always lower than the pH${}_{e}$, especially as time increases (Figure 2A). For this cell line, there is no inversion of the pH gradient as usually documented in tumor cells [46]. In addition, a quasilinear increase in pH${}_{i}$ with pH${}_{e}$ in the range of [5.0–7.2] is observed, and the values of pH${}_{i}$ are very close to those of pH${}_{e}$. Moreover, the dependence of pH${}_{i}$ as a function of time is low in the acidic pH${}_{e}$ range, as the curves are almost superimposed. However, the range with higher pH${}_{e}$ values shows a marked dependence of the pH${}_{i}$ over time. In the first 4 h, the pH${}_{i}$ of F98 cells decreases sharply and stabilizes after 5 h. At that time, the relationship between pH${}_{i}$ and pH${}_{e}$ exhibits a plateau for pH${}_{e}$ values above 7.2. Therefore, F98 cells have the ability to progressively adjust their pH${}_{i}$ over time under basic conditions (above 7.2).

#### 3.1.2. pH Changes in U87 Cells

For U87 cells, in the pH${}_{e}$ range of [5.0–6.8], pH${}_{i}$ is higher than pH${}_{e}$ which, in agreement with the reverse pH gradient described in tumor cells (Figure 2B) [46]. Therefore, U87 cells are able to increase their pH${}_{i}$ under acidic conditions. A time dependence of pH${}_{i}$ is also noted in this pH${}_{e}$ range. pH${}_{i}$ increases over 3 h and is less influenced after 4 h. However, in the zone of pH${}_{e}$ > 6.8, pH${}_{i}$ is always less than pH${}_{e}$, and pH${}_{i}$ is completely independent of time (superimposed curves). Therefore, U87 cells have the ability to rapidly maintain a pH level close to the physiological level for pH${}_{e}$ values above 6.8.

#### 3.1.3. Interpretation

In their basal state, i.e., at the physiological pH of the culture medium, F98 cells have a pH${}_{i}$ of 7.65 ± 0.13. Lowering the pH${}_{e}$ to 5.00 rapidly decreases the pH${}_{i}$ to 4.94 ± 0.08 (after 1 h). This decrease of pH${}_{i}$ corresponds to an increase in hydrogen ions (H${}^{+}$) inside F98 cells.

The main mechanism responsible for pH${}_{i}$ changes is the movement of H${}^{+}$ ions across the plasma membrane [47,48,49]. Extracellular acidification stimulates the passive influx of H${}^{+}$ ions into the cell by increasing the permeability of the plasma membrane, resulting in a decrease in pH${}_{i}$ [50]. As acidic pH${}_{e}$ limits the availability of bicarbonate ions (HCO${}_{3}^{-}$) and thereby reduces both passive and dynamic HCO${}_{3}^{-}$-dependent buffering, tumor cells are thought to use a sodium–hydrogen exchanger (NHE exchanger) to recover from intracellular acidification. However, their poor recovery during this assay suggests that NHE activity is low in F98 glioma cells, in agreement with the study by Glunde et al., 2002 [51], who showed that extracellular acidification inhibits the activity of NHE-1 exchanger (sodium–hydrogen exchanger isoform-1) in glioma cell lines (C6 and F98).

For basic pH${}_{e}$, F98 cells maintain a physiological level of pH${}_{i}$ around 7.4 (more precisely, 7.33 ± 0.13). Changing from an initial state of pH${}_{i}$ 7.65 ± 0.13 to 7.33 ± 0.13, a passive influx of H${}^{+}$ ions into the cell is imposed. Although basic pH${}_{e}$ limits the availability of H${}^{+}$ ions in the culture medium, H${}^{+}$ ions are generated by the hydration of CO${}_{2}$ according to the following reaction:(1)$${CO}_{2}+H_{2}O⇌H_{2}{CO}_{3}⇌H^{+}+{HCO}_{3}^{-}$$

The H${}^{+}$ ions entering the cells are sequestered by the bicarbonate ions (HCO${}_{3}^{-}$), increasing the pH${}_{i}$. However, this is not the case with F98 cells. As F98 cells have reduced their pH${}_{i}$, it can be assumed that they use the Cl${}^{-}$/HCO${}_{3}^{-}$ exchanger to export the HCO${}_{3}^{-}$ ions outside the cells. Once the HCO${}_{3}^{-}$ ions are exported, H${}^{+}$ ions are no longer sequestered, leading to a decrease in pH${}_{i}$. The Cl${}^{-}$/HCO${}_{3}^{-}$ exchanger has been identified as an essential regulator particularly activated following alkaline incubation [52,53,54]. Activation of the Cl${}^{-}$/HCO${}_{3}^{-}$ exchanger decreases the rate of acidification during recovery of pH${}_{i}$ from extracellular alkaline incubation.

In their basal state, U87 cells have a pH${}_{i}$ of 7.48 ± 0.12. Lowering the pH${}_{e}$ to 5.00 decreases pH${}_{i}$ to 4.75 ± 0.14 (after 1 h). The rapid drop in pH${}_{i}$ is due to the passive influx of H${}^{+}$ ions into the cell. However, unlike F98 cells, U87 cells have a reduced intracellular acidity of 5.81 ± 0.12 after 8 h. Therefore, considering the low-HCO${}_{3}^{-}$-dependent buffering at acidic pH levels, it can be suggested that U87 cells use the dynamic NHE exchanger to recover from intracellular acidification. The plasma membrane isoforms of this protein extrude one intracellular H${}^{+}$ ion in exchange for one extracellular Na${}^{+}$ ion, leading to an increase in pH${}_{i}$.

For basic pH${}_{e}$, U87 cells maintain a pH${}_{i}$ level close to physiological pH (more precisely, 7.25 ± 0.03). Changing from an initial state of pH${}_{i}$ 7.48 ± 0.12 to 7.25 ± 0.03, like F98 cells, U87 cells use the Cl${}^{-}$/HCO${}_{3}^{-}$ exchanger to export the HCO${}_{3}^{-}$ ions outside the cells and to recover from alkaline incubation by decreasing pH${}_{i}$. The temporal dependence of pH${}_{i}$ observed in F98 cells can be explained by the fact that the pH${}_{i}$ of F98 cells at time zero is higher than that of U87 cells (by about 0.2 pH units).

### 3.2. Resistance to Extracellular pH Changes

Many recent studies have shown that cancer cells exhibit a higher level of resistance to acidity due to their ability to regulate their pH${}_{i}$ [55,56,57]. As the two investigated cell lines do not regulate their acidity in the same way, we were interested in evaluating the effects of acidification of the tumor microenvironment on the mortality of those glioma cells.

Cell resistance to instantaneous external pH changes was assessed by measuring cell mortality every 4 h for 72 h, i.e., until the mortality level was higher than 90% for all tested experimental conditions. The measurements were performed according to the protocol described in the Materials and Methods section. Two graphical representations of the results are presented. The first shows the evolution of the percentage of dead cells (relative to the total number of cells in the population) over time (Figure 3A,B). The second represents the excess mortality compared to the control (which is standard culture medium with a physiological pH of 7.4) (Figure 3C,D). These curves were obtained by subtracting the control from each curve in Figure 3A,B, with the control now set to 100%. We note that if the excess mortality is lower than 100%, then the mortality is reduced compared to the control condition.

Results are shown in Figure 3. Both cell lines are resistant to highly acidic conditions over 48 h, with at least 15% of cells surviving. After 72 h, the cells are almost all dead (Figure 3A,B). At 28 h, F98 mortality is higher under the initial extreme acidic (4.00) and basic (9.00) pH${}_{e}$ conditions, with a mortality of approximately 53% and 60%, respectively. However, U87 mortality is higher for under acidic pH${}_{e}$, with cell mortality of approximately 60%, 54%, and 51% under the initial pH${}_{e}$ of 4.00, 5.00, and 6.00, respectively. Beyond 28 h, cell mortality becomes more important, and compared to the control state, the excess mortality reaches nearly 125% for certain pH${}_{e}$ values (Figure 3C,D). F98 cells appear to have a higher level of resistance to acidity than U87 cells, whereas U87 cells have a higher level of resistance to basicity than F98 cells.

Several studies have reported the existence of a link between the regulation of intracellular acidity and the reduction in cell death [58,59,60,61]. Therefore, in order to minimize toxic intracellular acidity, tumor cells must regulate the extrusion mechanisms of H${}^{+}$ ions, which increase pH${}_{i}$. Taking these considerations into account, U87 cells that regulate intracellular acidity should exhibit lower cell mortality than F98 cells that do no regulate intracellular acidity. However, that is not the case here. U87 cells exhibit higher levels of mortality under acidic conditions. In this case, the regulation of intracellular acidity does not seem to constitute an adaptive response, since the regulation of acidity does not protect from death. The pH regulation process may be too costly to the cell.

### 3.3. Intracellular pH in Spheroids

In this section, experiments on spheroid are considered. Their 3D structure leads to spatial heterogeneities [62] associated with the limited diffusion of oxygen and nutrients from the periphery to the core [63,64]. Therefore, the cells of a spheroid experience a radial gradient of oxygen often associated with a reverse radial gradient of acidity, since hypoxic cells are the source of acidity. Moreover, contrary to the 2D culture conditions, cells in a spheroid are in close contact, and cell–cell binding interactions influence the ability of the cells to adapt to the extracellular environment and to resist death. Therefore, we assessed pH${}_{i}$ spatial heterogeneity in this 3D context for the two cell lines.

Under identical culture conditions, F98 spheroids (diameter of 430–490 μm) are larger than U87 spheroids (diameter of 290–330 μm) (Figure 4). It has been reported that the size of spheroids, ranging from tens to hundreds of microns in diameter, is critical for cellular functions and model applications; thus, spheroids of different sizes do not exhibit the same properties [65]. For example, spheroids with a diameter < 150 μm do not develop chemical gradients (pH${}_{i}$, oxygen concentration, etc.). These gradients are usually generated in spheroids with diameters > 200 μm and increase with increasing spheroid size [65,66]. Therefore, due to a size-induced oxygen difference, larger pH${}_{i}$ gradients are expected in F98 spheroids than in U87 spheroids. Moreover, F98 spheroids are much more spherical than U87 spheroids with a denser and more compact shape. Therefore, the nature of cell–cell interactions probably differs between the two cell lines.

Figure 5 shows the pH${}_{i}$ maps for a slice located at a depth of 85 μm in F98 and U87 spheroids. The pH maps exhibit a pH${}_{i}$ gradient with the lowest value at the center of the slice. F98 spheroids, which are larger than U87 spheroids, exhibit a more irregular pH${}_{i}$ gradient than U87 spheroids, which are smoother”. These gradients are directly correlated with the depletion of oxygen from the periphery to the core of the spheroid, since the pH${}_{i}$ values measured at the center are systematically lower than the pH${}_{e}$ values (taken at 8 h). We recall that the DMEM undergoes a pH drift, with values summarized in Table 1.

For a quantitative analysis, the pH${}_{i}$ maps (Figure 5) were processed in two ways. First the distribution of the pH${}_{i}$ measured for each pixel as a function of the pixel distance from the center of the spheroid was plotted (Figure 6). The added value of this representation is to retain the spatial information contained in the pH${}_{i}$ map. Second, the pH${}_{i}$ maps of the spheroids were segmented to delineate three zones: a central zone, an intermediate zone, and a peripheral zone (Figure 7). A mean pH${}_{i}$ value was associated with each zone. The following two paragraphs are dedicated to these two types of analysis.

#### 3.3.1. pH Distribution

The distribution of the pH${}_{i}$ measured for each pixel as a function of the pixel distance from the center of the spheroid is represented in Figure 6 for the two cell lines exposed to three different initial pH${}_{e}$ values (5.00, 6.00 and 7.40). Each point on the pH${}_{i}$ distribution represents a pixel of the map, with a pixel size of 1.24 μm for F98 spheroids and 0.83 μm for U87 spheroids. Pixel sizes are much smaller than the size of a cell; as a consequence, cellular heterogeneity can be assessed, since a cell has pH${}_{i}$ values that spread over several pixels. The color code of the pH${}_{i}$ distribution indicates the density of superimposed pixels.

The pH${}_{i}$ distributions of F98 spheroids (Figure 6, left) show a wide dispersion of points. However, this dispersion becomes less important approaching the center and the periphery. The majority of cells located at a distance ranging from 100 μm to 200 μm from the center have a pH${}_{i}$ in the range of [5.00–5.50], [5.50–6.50], and [8.00–9.00] when incubated at initial pH${}_{e}$ values of 5.00, 6.00, and 7.40, respectively. However, by following the evolution of pH${}_{i}$ as moving away from the center of the spheroid, we can see that a large group of cells tends to have pH${}_{i}$ values above 5.50 and 6.50 with initial pH${}_{e}$ values of 5.00 and 6.00, respectively. On the other hand, and under basic conditions, although the majority of F98 cells located near the center of the spheroid have a pH${}_{i}$ in the range of [8.00–9.00], a small group of cells tends to have pH${}_{i}$ values of less than 8.00. It appears that the F98 cells located around 150 μm from the center of the spheroid attempt to regulate their pH${}_{i}$ towards the physiological state.

The pH${}_{i}$ distributions of U87 spheroids (Figure 6, right) show a considerably lower dispersion of points than those of F98 cells, with lines of higher density with increasing pH moving away from the center of the spheroid. Hence, the distribution of pH${}_{i}$ is much more homogeneous in the U87 spheroids. In addition, depending on the point density, the majority of points have a pH${}_{i}$ in the range of [4.00–4.50], [5.00–6.00], and [6.00–6.00] when incubated at initial pH${}_{e}$ values of 5.00, 6.00, and 7.40, respectively. Therefore, U87 cells do not regulate their pH${}_{i}$ and are more acidic than F98 cells in a spheroid.

It has been reported that pH${}_{i}$ depends on the adhesive interactions of cells with neighboring cells and that cell–cell contact interactions regulate cell activities via modulation of pH${}_{i}$ [67,68]. The fact that central F98 cells tend to regulate their pH${}_{i}$ may be associated with an increase in the number of cell–cell contacts inside the spheroid.

Since tumor spheroids represent heterogeneous 3D structures in terms of pH${}_{i}$ in which the cells of the outer proliferative layer have a more alkaline pH${}_{i}$ than the quiescent cells inside [69], pH${}_{i}$ measurements were performed separately for the inner and outer layers of the spheroids. The pH${}_{i}$ maps presented in Figure 5 were used to produce a spheroid outline. This contour was used to generate three concentric, non-overlapping layered regions of interest (ROIs) with a width equal to one-third of the spheroid radius (Figure 7). ROI${}_{c}$, ROI${}_{m}$, and ROI${}_{p}$ are defined as the center, intermediate, and peripheral ROI, respectively. Therefore, a mean pH${}_{i}$ value is associated with each zone. Radial pH${}_{i}$ data are summarized in Figure 8.

#### 3.3.2. pH${}_{i}$ in ROI

Analysis of pH${}_{i}$ separately in the core and on the periphery of the F98 spheroids showed a significant difference between the central and peripheral parts (Figure 8A). Quantitative analysis of the pH${}_{i}$ values revealed that the pH${}_{i}$ in the central areas was 5.08 ± 0.21, 5.87 ± 0.28, and 7.51 ± 0.31 for spheroids incubated at initial pH${}_{e}$ values of 5.00, 6.00, and 7.40, respectively, whereas the pH${}_{i}$ in peripheral parts of these spheroids was 5.97 ± 0.21, 7.18 ± 0.33, and 8.57 ± 0.27, respectively. Therefore, F98 spheroids showed 0.89, 1.31, and 1.06 units of difference in pH${}_{i}$ between the center and the periphery when incubated at initial pH${}_{e}$ values of 5.00, 6.00, and 7.40, respectively. Therefore, regardless of the pH${}_{e}$, F98 spheroids develop significant radial gradients of pH${}_{i}$, with the lowest levels reached at the core.

In U87 spheroids, quantitative analysis of the pH${}_{i}$ values revealed that the mean pH${}_{i}$ in the central areas was 4.17 ± 0.16, 5.72 ± 0.04, and 6.06 ± 0.08 for spheroids incubated at initial pH${}_{e}$ values of 5.00, 6.00, and 7.40, respectively, (Figure 8B), whereas the pH${}_{i}$ in peripheral parts of these spheroids was 4.41 ± 0.25, 6.03 ± 0.09, and 6.65 ± 0.20, respectively. Therefore, U87 spheroids showed 0.24, 0.31, and 0.59 units of difference in pH${}_{i}$ between the center and the periphery when incubated at initial pH${}_{e}$ values of 5.00, 6.00, and 7.40, respectively. According to another observation, the gradient of pH${}_{i}$ decreases with decreasing pH${}_{e}$, and the distribution becomes quasihomogeneous under very acidic conditions.

To compare the pH${}_{i}$ gradients between the F98 and U87 spheroids, the slope of each gradient was calculated (Figure 8). F98 spheroids show a pH${}_{i}$ gradient with a slope of 0.53, 0.66, and 0.44 when incubated at a pH${}_{e}$ of 5.60, 6.60, and 8.20, respectively, while the U87 spheroids show a pH${}_{i}$ gradient with a slope of 0.29, 0.15, and 0.12 when incubated at a pH${}_{e}$ of 5.60, 6.60, and 8.20, respectively. For a given pH${}_{e}$, U87 spheroids consistently present pH${}_{i}$ gradients with a smaller slope than those obtained for the gradients of F98 spheroids. This verifies what we expected based on the greater size and density of F98 spheroids.

#### 3.3.3. 2D vs. 3D Cultures

In order to compare the results obtained in 2D cultures and 3D spheroids, the relationship between pH${}_{e}$ and pH${}_{i}$ obtained in these two types of culture (8 h after incubation at a given pH${}_{e}$) was plotted on the same graph (Figure 9).

Under acidic and neutral conditions, the peripheral F98 cells show a response similar to that of the cells in 2D cultures. Under basic conditions, peripheral cells tend to have increased pH${}_{i}$ values compared to the 2D model. Having already assumed that under basic conditions, the F98 cells in the 2D cultures use the Cl${}^{-}$/HCO${}_{3}^{-}$ exchanger to export the HCO${}_{3}^{-}$ ions outside the cells, we can hypothesize that the activity of the Cl${}^{-}$/HCO${}_{3}^{-}$ exchanger is inhibited in 3D cultures.

However, regardless of the pH${}_{e}$, the U87 cells show a completely different activity compared to the 2D cultures. In the spheroid, U87 cells tend to have lower pH${}_{i}$ values compared to the 2D model. We have already assumed that under acid conditions, the U87 cells in the 2D cultures use the Na${}^{+}$/H${}^{+}$ exchanger to export the H${}^{+}$ ions outside the cells. The fact that under acidic conditions, the U87 cells lowered had reduced intracellular pH can be explained by the inhibition of this exchanger, thereby conserving H${}^{+}$ ions in the cells. However, unlike F98 cells, the decrease in pH${}_{i}$ under basic conditions allows us to hypothesize that U87 cells “overexpress” Na${}^{+}$/H${}^{+}$ in the spheroid configuration.

### 3.4. Effect of pH on Temozolomide Efficacy

Temozolomide (TMZ) is currently the standard treatment for glioblastoma (GBM), since it is one of the rare molecules able to cross the blood–brain barrier. However, there are serious drawbacks, mainly related to the invasive nature of this tumor and to inherent and acquired resistance, which can ultimately lead to treatment failure. Therefore, there is an urgent need for novel therapeutic strategies that enhance the benefits of TMZ in terms of patient survival and quality of life.

The effect of TMZ is highly pH-dependent [70]. TMZ is stable under acidic conditions. It is converted into its active form in a two-step process under physiological and basic pH conditions. However, since tumor cells can acidify their microenvironment, this may influence the efficacy of TMZ. Therefore, only a small pH range favors TMZ-induced damage in tumor cells [35]. In this context, the influence of the extracellular pH (pH${}_{e}$) on the efficiency of the TMZ complex was studied here in order to determine the optimal pH${}_{e}$ for its use on the two cell lines of interest, i.e., F98 and U87.

The majority of previously published drug toxicity research was conducted using commercially available cell media with a pH above 7, which is typical of healthy tissues. These experiments do not reflect the actual conditions of pathological cell division and growth. Therefore, to perform biological experiments under conditions similar to those in cancer cells, it is important to adjust the pH of cell media to the appropriate value. The experiments reported in this work were carried out in neutral and acidified cell media, and the influence of pH on the toxicity results in the U87 and F98 cell lines was evaluated.

Figure 10 shows the dependence of TMZ toxicity on the pH${}_{e}$ of the medium itself 48 h after exposure for the two cell lines in 2D cell cultures and in spheroids. We recall that the pH${}_{e}$ is subjected to a drift over the 48 h, with values summarized in Table 1. The results show that in most of the cases, there is no significant difference between cultures with DMSO alone and cultures without DMSO.

Table 2 summarizes the results presented in Figure 10. The cell viability was evaluated in each case by comparison with the control case with an initial physiological pH${}_{e}$ = 7.4 with DMSO. For values above 100%, there is an enhanced viability compared to the control case. This occurs in untreated F98 cells in 2D cultures at initial an pH${}_{e}$ = 7.2 and in untreated spheroids for both cell lines at an initial pH${}_{e}$ = 6.

In order to estimate the optimum pH${}_{e}$ for TMZ efficacy, the cell viability gap, which measures the difference between the viability of control and TMZ-treated cells, was calculated. The greater the gap, the more effective the therapy. The results are presented in Figure 11.

For both cell lines in 2D cell cultures (Figure 11A,B), the toxicity of TMZ decreases with decreased pH${}_{e}$ values, and a pH${}_{e}$ of 8.52 provides a satisfactory efficiency of the drug. Interestingly, F98 cells exhibit higher sensitivity toward TMZ treatment at an initial pH${}_{e}$ value of 6.80 compared to U87. The cell viability is 47 points for F98 cells and 21 points for U87 cells. For the other pH${}_{e}$ values, whether lower or higher, both cell lines behaved similarly. U87 and F98 cells show low sensitivity to TMZ at initial pH${}_{e}$ values of 6.00 and 6.40, with viability gaps below 25 points. For both cell lines, the optimal pH${}_{e}$ for TMZ efficiency is obtained with an initial pH${}_{e}$ of 7.40 and a drift to 8.52 over the 48 h of treatment. Since the pH drift usually occurs within a few hours and assuming that a cell viability gap above 40 points is sufficiently significant, the optimal pH${}_{e}$ range for TMZ efficiency is [7.67–8.52] for F98 cells and [8.15–8.52] for U87 cells.

#### Toxicity in Spheroids

Figure 12 presents fluorescent images of F98 and U87 spheroids exposed to a change in pH${}_{e}$ after 3 days of growth. Spheroids were then exposed to DMSO only or TMZ for 48 h. The images show that TMZ-treated spheroids are smaller at all initial pH${}_{e}$ values for the F98 cells (Figure 12A). On the other hand, U87 cells appear to be less sensitive to TMZ, and the size reduction appears to increase as the initial pH${}_{e}$ increases (Figure 12B).

The cell viability gap, i.e., the difference between the viability of control and TMZ-treated cells, shows a pH-dependent increase towards higher pH value for both cell lines. As expected from the fluorescence images, the TMZ toxicity is more marked for F98 cells than U87 cells, with viability gaps of 60 and 70 points for F98 spheroids (Figure 11C) versus 43 and 63 points for U87 spheroids (Figure 11D) at initial pH${}_{e}$ values of 6.00 and 7.40, respectively. For the lowest initial pH${}_{e}$ of 5.00, the efficiency of TMZ is reduced for both cell lines, since TMZ is not activated under acidic conditions.

If we compare TMZ efficacy in 2D cultures and 3D spheroids, a discrepancy is observed at an initial pH${}_{e}$ values of 6.00. For both cell lines, the TMZ efficacy is limited in 2D cultures and significant in spheroids. In 2D cultures, the cells are homogeneously exposed to a pH${}_{e}$ evolving from 6.00 to 6.91 due to pH drift. This acidic pH tends to maintain the TMZ in its inactive form. On the other hand, in spheroids, relatively good efficacy of TMZ is observed, with a cell viability above 40 and a higher level of mortality for F98 cells. Considering the pH${}_{i}$ gradients in Figure 5, we observe that the intracellular pH can reach values close to basicity (as the pH${}_{e}$ drift is higher at 48 h). The TMZ effect is more important in F98 cells, which is coherent with the fact that the spheroids are larger and more compact, which gives rise to a more spread pH${}_{i}$ gradient, with a population of cells presenting TMZ-sensitive pH values. On the other hand, U87 spheroids are smaller and much more homogeneous in terms of cell pH${}_{i}$, and a smaller population is TMZ-sensitive.

Overall, the results show that the efficacy of TMZ is modulated by the cell type, suggesting the possibility of using pH${}_{e}$ as a personalized therapeutic target.

## 4. Discussion

Tumor metabolism has attracted renewed attention recently, as the acidity generated by tumor cells has been found to be responsible for increased cell aggressiveness, favoring invasion and resistance to treatment [71]. The true nature of cancer cell metabolism has become an active source of debate with respect to whether it should be considered an evolutionary specialization, an atavistic remanence [71,72], or a contextual adaptation to extreme conditions [73,74,75].

In order to survive the dynamic nature of pH in tumors, cancer cells require the ability to sense tiny pH changes and respond appropriately to maintain pH${}_{i}$ homeostasis, i.e., the ability to regulate pH${}_{i}$. Regulation of pH${}_{i}$ starts with changes in the expression or activity of several plasma membrane molecules, such as pumps and transporters, which facilitate proton efflux. Thus, a strength in understanding the pH${}_{i}$ regulation of tumors is the understanding of proton transport across the plasma membrane. Therefore, we experimentally characterized the pH${}_{i}$-regulating capacity of two glioma cell lines in monolayer cultures using fluorescence microscopy. We observed that the pH${}_{i}$ regulation mechanism is not the same for the two cell lines. First, under acidic conditions, U87 human glioblastoma cells are able to reduce intracellular acidity by exporting protons outside the cells. In other words, U87 cells are able to regulate acidic pH. We assumed that U87 cells use the NHE-1 exchanger to export protons in exchange for Na${}^{+}$ ions. However, we found that F98 cells are not able to regulate acidic pH, with H${}^{+}$ ions maintained inside cells. Therefore, another mechanism might be involved, since high pH${}_{i}$ values degrade intracellular proteins and prevent normal functioning of the cell machinery. On the other hand, we found that both cell lines are able to regulate basic pH. We hypothesized that the cells use the Cl${}^{-}$/HCO${}_{3}^{-}$ exchanger to export HCO${}_{3}^{-}$ ions outside the cells, maintaining a pH${}_{i}$ level close to the physiological level.

F98 cells that do not regulate intracellular acidity preserve protons inside the cells. In turn, the intracellular protons produce a toxic effect on many cellular processes, such as enzyme activities, metabolism, and gene expression, leading to cell death [76,77]. However, this is not the case with F98 cells. F98 cells showed a high resistance to intracellular acidity. Accumulating evidence suggests that in addition to the well-characterized ion pumps and exchangers in the plasma membrane, cancer cell lysosomes help to avoid potential toxic acidification of the intracellular milieu [78,79,80]. The lysosome is an intracytoplasmic organelle responsible for the degradation and recycling of intra- or extracellular components. It is very heterogeneous in terms of size, contains 50 to 60 hydrolases (phosphatases, glycosidases, lipases, nucleases, sulfatases, and proteases), and is active at acidic pH. It has been reported that the pH of lysosomes is maintained at 4.5–5 and that the concentration of H${}^{+}$ ions in lysosomes is almost 1000 times higher than that in the cytosol [80,81]. Therefore, the lysosome is not only a place for protein degradation but also a storage compartment for H${}^{+}$ ions. The pH of lysosomes is regulated by V-ATPase, a pump located in the membrane of these organelles, which brings H${}^{+}$ ions into these vesicles and acidifies the contents. We suggest that H${}^{+}$ ions are encapsulated in F98 cells and therefore assume that F98 cells activate the V-ATPase pump at acidic pH to bring H${}^{+}$ ions into the lysosomes.

In Figure 13, we present a graphical summary of our hypotheses we make based on the corpus of knowledge available in the literature about the mechanisms possibly involved in the regulation of intracellular acidity for the two investigated cell lines. We aim to provide an interpretation of the results we obtained, but these hypotheses remain to be verified.

U87 cells, which were found to regulate intracellular acidity, use the NHE exchanger to export H${}^{+}$ ions outside the cells. Fluxes through the NHE are driven only by the combined chemical gradients of Na${}^{+}$ and H${}^{+}$ and, therefore, do not directly consume metabolic energy [82]. However, it has been found that the presence of physiological levels of ATP in certain cell lines is necessary for optimal Na${}^{+}$/H${}^{+}$ exchange [21,83,84,85]. As already seen, U87 cells were able to reduce intracellular acidity through Na${}^{+}$/H${}^{+}$ exchange over 8 h. The exchanger operated normally during this period. Assuming that the activity of an NHE exchanger in U87 cells depends on the availability of intracellular ATP, the NHE exchanger might have exhausted the intracellular ATP necessary for the various mitochondrial functions, in particular for maintaining the membrane potential gradient, which stimulates cell survival and proliferation [86,87]. Therefore, the depletion of ATP by the NHE exchanger leads to cell death, which may explain the higher mortality of U87 cells compared to F98 cells.

Very few studies have been conducted to investigate the effect of alkaline pH${}_{e}$ on cell survival [88,89,90,91,92]. These studies did not focus on the cellular metabolism but rather on the effect of extracellular alkalinization on the permeability of the plasma membrane [88,91,92]. It was found that the morphology of cells was altered in alkalinized culture. Spindle-shaped cells, like F98 cells, formed small patches of round cells at basic pH${}_{e}$ [92]. Moreover, the cells lost their adhesion ability. Consequently, the membrane potential of mitochondria was altered, leading to permeability transition, pore opening, and cell death [93,94]. Cells that grow as cell aggregates, such as U87 cells, are therefore more resistant to cell death under basic pH${}_{e}$.

### Conclusions and Future Work

Many cellular processes and therapeutic agents are known to be highly pH-dependent, making the study of intracellular pH (pH${}_{i}$) regulation of utmost importance. Therefore, this study made it possible to focus on the regulation of the pH${}_{i}$ of two brain tumor cells using fluorescence microscopy. We observed that the tumor regulation of acidity is not the same for the two cell lines; as a consequence, our results do not support the common idea that tumor cells behave in a similarly way. On the other hand, pH${}_{i}$ regulation appears to be highly cell-dependent.

In order to interpret our results, we proposed hypotheses to explain the mechanisms involved in the regulation of pH${}_{i}$. In 2D monolayer cultures, we assumed that F98 rat glioma cells do not regulate intracellular acidity and preserve protons inside the cells by activating the V-ATPase pump at acidic pH to bring H${}^{+}$ ions into lysosomes. A study of the effect of the inhibition of the V-ATPase pump by bafilomycin A1 [95] on pH${}_{i}$ would be an excellent tool to validate our interpretation. On the other hand, we hypothesized that human glioblastoma U87 cells, which were found to regulate intracellular acidity, use the Na${}^{+}$/H${}^{+}$ exchanger to export H${}^{+}$ ions outside the cells. Therefore, testing amiloride as a Na${}^{+}$/H${}^{+}$ exchanger inhibitor [96] may be an effective tool to validate our hypothesis.

In addition, in order to properly characterize the tumor regulation of pH${}_{i}$, it is crucial to study it at the scale of a spheroid. Such a 3D model constitutes an intermediate biological model between 2D cell cultures and animal models since a relationship is developed between the cells and their heterogeneous environment characterized by substrate fluxes and gradients. More precisely, cells growing in a spheroid are exposed not only to reduced pH${}_{e}$ levels, as in 2D monolayer cultures, but also to a limitation of the supply of nutrients and oxygen due to their diffusion inside the spheroid. Therefore, differences in results between 2D and 3D experiments were expected. Tidwell et al., 2022 [97] recently showed that the combination of hypoxia, starvation of other nutrients, and extracellular acidity inhibits energy metabolism and protein synthesis and prevents the regulation of exchanges such as Na${}^{+}$/H${}^{+}$ in 3D spheroids, in agreement with the results obtained in U87 spheroids and contrary to the results obtained in 2D monolayer cultures. The fact that the U87 spheroids had reduced pH${}_{i}$ under acidic conditions can be explained by the inhibition of the Na${}^{+}$/H${}^{+}$ exchanger.

The effect of temozolomide (TMZ), the cornerstone drug used to treat brain tumors, is highly pH-dependent [98]. Therefore, the fact that tumor cells can acidify their microenvironment and regulate their intracellular pH may influence the efficacy of TMZ. A recent theoretical study showed that the pH${}_{i}$ regulation capacity of a cell line can be exploited to estimate the optimum pH${}_{e}$ for TMZ efficacy [35]. Therefore, the effect of TMZ was studied on our two cell lines by manipulating the extracellular pH (pH${}_{e}$). The results allowed us to show that drug efficiency depends on the cell type and on the pH${}_{e}$, suggesting the use of pH as a personalized therapeutic target for future research based on the combination of TMZ and pH-regulating agents.

## Figures and Tables

**Figure 1 biology-12-01221-f001:**
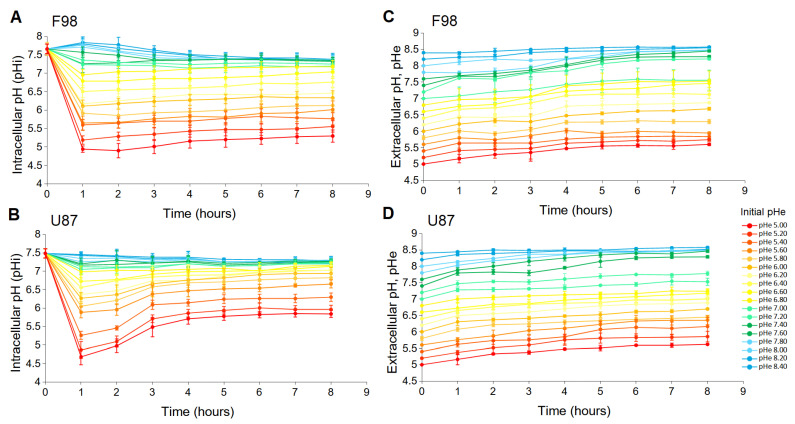
Evolution of pH${}_{i}$ with sudden changes in the extracellular pH from 5 to 8.4 for F98 cells (**A**) and U87 cells (**B**). Simultaneous evolution of the pH${}_{e}$ of the culture medium of F98 cells (**C**) and U87 cells (**D**). Each measured point represents the mean (±SD) of three independent experiments.

**Figure 2 biology-12-01221-f002:**
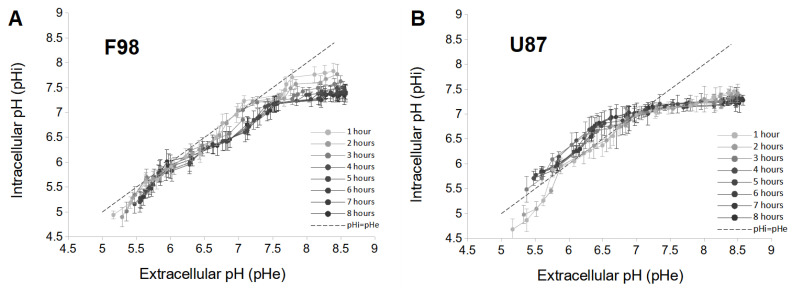
Relationship between pH${}_{e}$ and pH${}_{i}$ of F98 (**A**) and U87 (**B**) cells plotted for each hour from 1 to 8 h. Each measure point represents the mean (±SD) of three independent experiments.

**Figure 3 biology-12-01221-f003:**
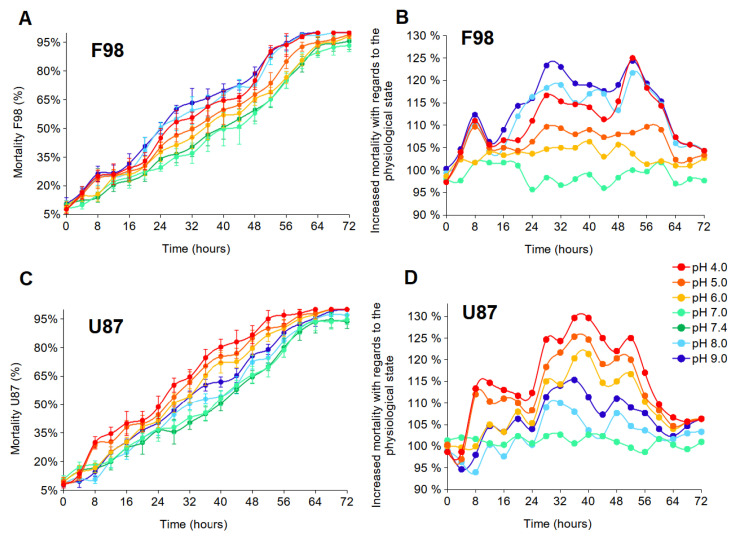
Effects of pH${}_{e}$ on cell mortality of F98 and U87 cells. (**A**,**B**) The evolution of the percentage of dead cells (relative to the entire number of cells in the population) over time. Each measured point represents the mean (±SD) of three independent experiments. (**C**,**D**) The excess mortality compared to the control set at 100% (which is standard culture medium with an initial physiological pH of 7.4).

**Figure 4 biology-12-01221-f004:**
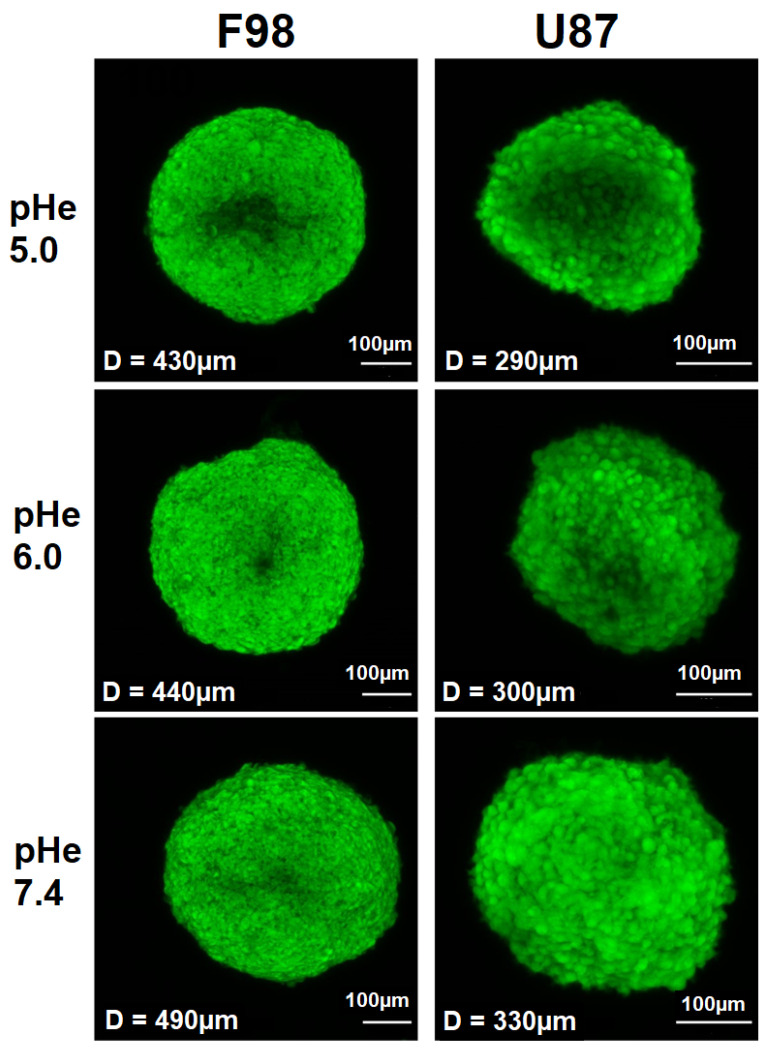
Fluorescent images of the spheroids generated by first collecting a stack of fluorescent images with a step size of 5 μm along the Z direction and projecting them to a high-quality 2D image by summing the slices. The spheroids were incubated at different initial pH${}_{e}$ values, and confocal Z-stack images were recorded 8 h after incubation. Scale bar = 100 μm.

**Figure 5 biology-12-01221-f005:**
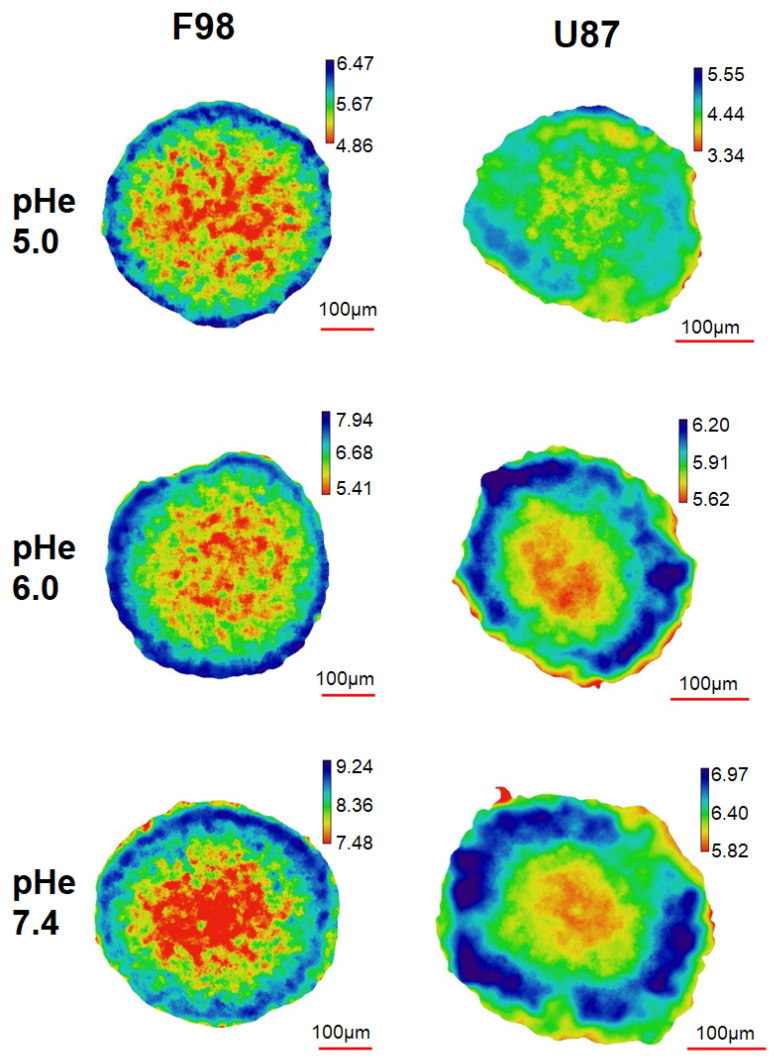
pH${}_{i}$ maps for a slice located at a depth of 85 μm for F98 (**left**) and U87 (**right**) spheroids. The spheroids were exposed to three different pH${}_{e}$ conditions for 8 h. Scale bar = 100 μm.

**Figure 6 biology-12-01221-f006:**
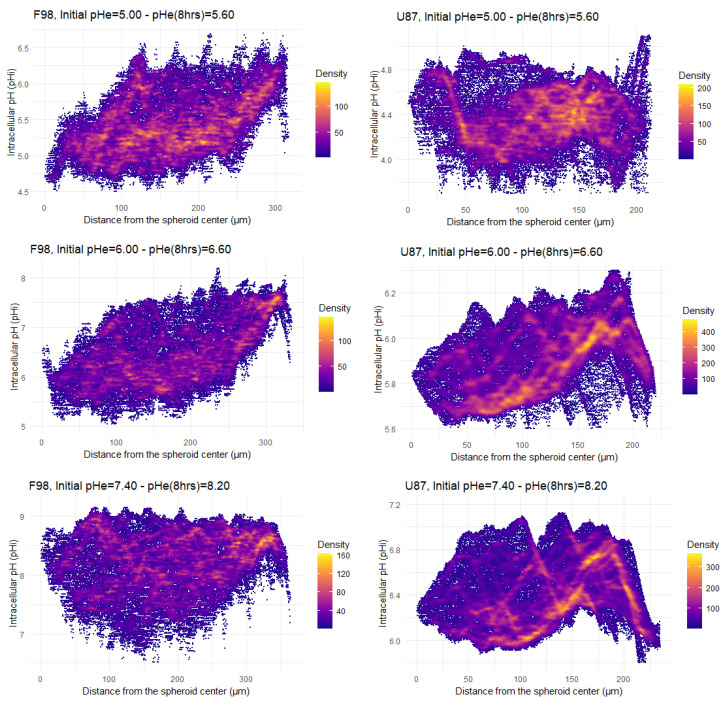
Radial profiles of pH${}_{i}$ distribution after 8 h of incubation of F98 (**left**) and U87 (**right**) spheroids. Each point corresponds to a pixel in the pH${}_{i}$ map; each pixel has an associated pH${}_{i}$ value and is located at a certain distance from the center of the spheroid. All the pixels are displayed in the pH${}_{i}$ maps.

**Figure 7 biology-12-01221-f007:**
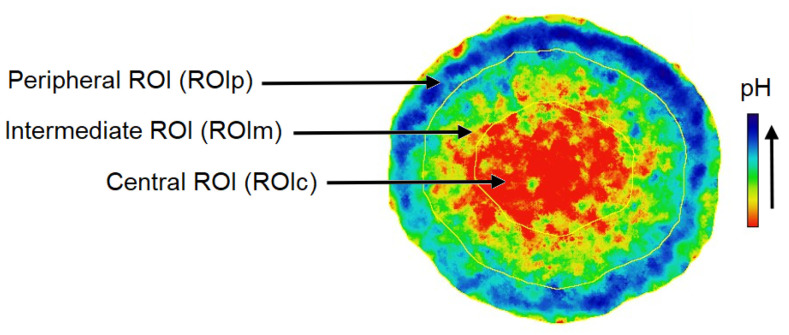
The contour of the spheroid was used to generate three regions of interest (ROIs) with a width equal to one-third of the spheroid radius. ROI${}_{c}$, ROI${}_{m}$, and ROI${}_{p}$ are defined as the center, intermediate, and peripheral ROI, respectively.

**Figure 8 biology-12-01221-f008:**
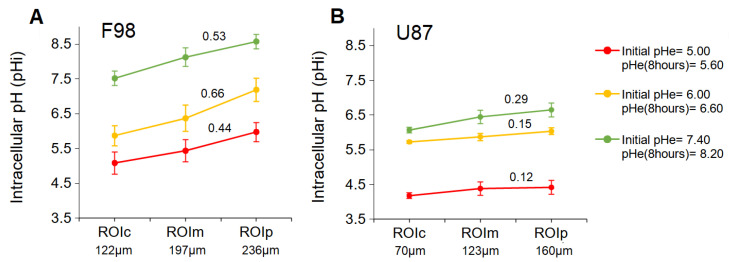
Mean pH${}_{i}$ gradient in F98 (**A**) and U87 (**B**) spheroids for three initial pH${}_{e}$ values as a function of the ROI from the center to the periphery.

**Figure 9 biology-12-01221-f009:**
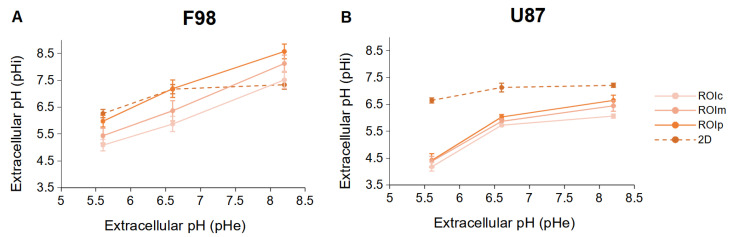
Relationship between pH${}_{e}$ and pH${}_{i}$ of F98 (**A**) and U87 (**B**) cells incubated in 2D cultures (dotted lines) and in the three ROIs of the spheroids (straight lines). The cells were incubated at different initial pH${}_{e}$ values, and the pH${}_{i}$ was measured 8 h after incubation. Each data point is the mean and SD of three replicates.

**Figure 10 biology-12-01221-f010:**
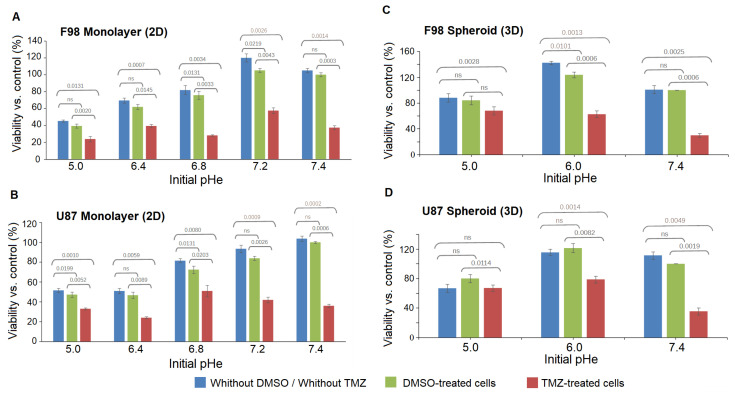
The dependence of TMZ toxicity on the initial pH${}_{e}$ of the medium itself for the F98 (**top**) and U87 (**bottom**) cell lines on 2D monolayer cultures (**A**,**B**) and on 3D spheroids (**C**,**D**). The control value is set at pH${}_{e}$ = 7.4 with DMSO. Measured points represent the mean (±SD) of three replicates. A statistical paired *t*-test was employed to compare the means of the viability of three biological repeats for each experimental condition at each initial pH${}_{e}$ value. Significant differences are defined as *p* < 0.05.

**Figure 11 biology-12-01221-f011:**
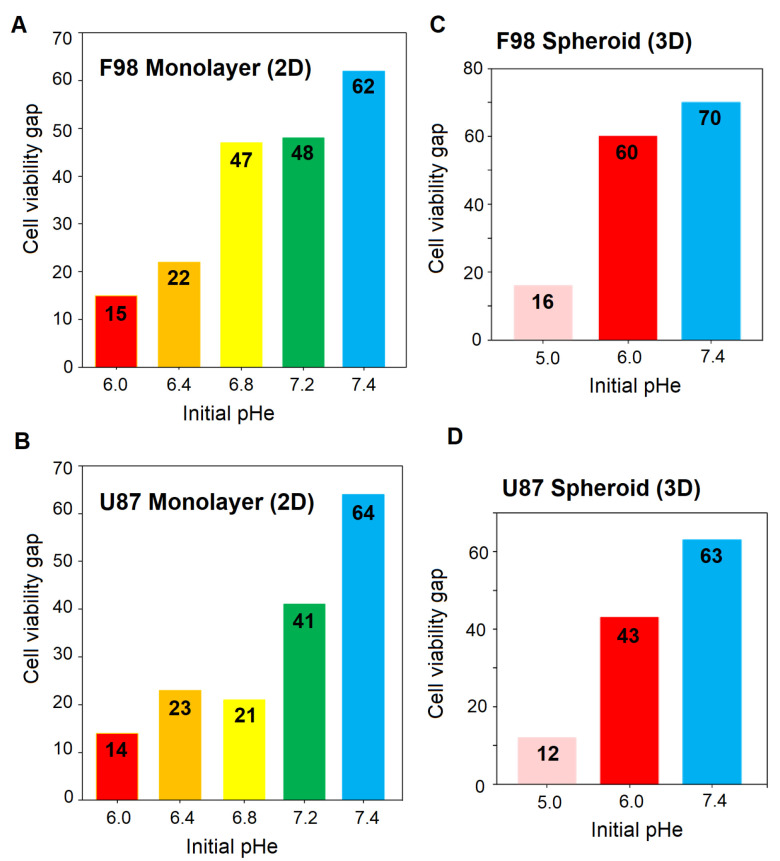
Cell viability gap after TMZ treatment of F98 and U87 for 2D monolayer cultures (**A**,**B**) and 3D spheroids (**C**,**D**). The cell viability gap is evaluated as the difference between the % cell viability before treatment (DMSO control) and the % cell viability after TMZ treatment.

**Figure 12 biology-12-01221-f012:**
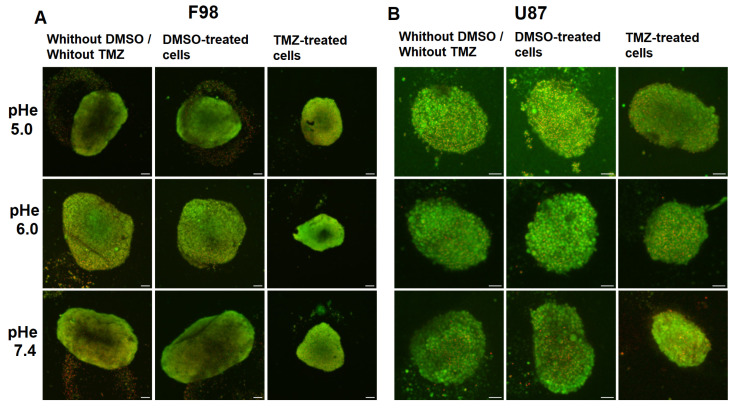
Fluorescent images showing the effect of TMZ on F98 spheroids (**A**) and U87 spheroids (**B**) under different initial pH${}_{e}$ conditions, without DMSO/without TMZ, with DMSO only, and with TMZ. Cells are marked in green with BCECF (pH${}_{i}$ indicator), and dead cells are marked in red with SRB. Scale bar = 100 μm.

**Figure 13 biology-12-01221-f013:**
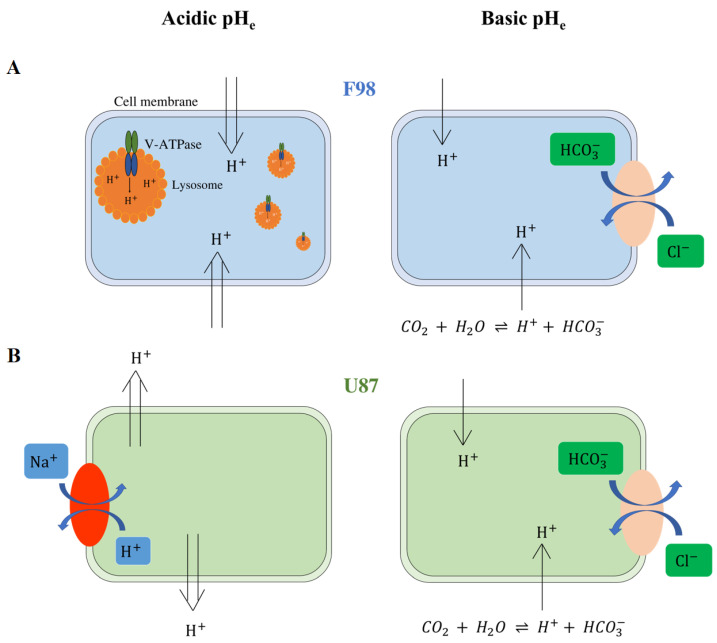
Hypotheses for the mechanisms of intracellular pH${}_{i}$ regulation of F98 (**A**) and U87 cells (**B**). F98 cells are not able to regulate acidic pH, and H${}^{+}$ ions remain inside the cells. To resist acidity, we hypothesize that F98 cells activate the V-ATPase pump to encapsulate the protons in lysosomes. On the other hand, U87 cells are able to regulate intracellular acidity by exporting protons outside the cells. We assume that U87 cells use the NHE-1 exchanger to export protons in exchange for Na${}^{+}$ ions. Both cell lines are able to regulate basic pH. We hypothesize that the cells use the Cl${}^{-}$/HCO${}_{3}^{-}$ exchanger to export the HCO${}_{3}^{-}$ ions outside the cells, maintaining a pH${}_{i}$ level close to the physiological level.

**Table 1 biology-12-01221-t001:** pH drift of DMEM after 8 h and after 48 h at 37 °C under 5% CO${}_{2}$.

Initial pH${}_{e}$	pH${}_{e}$ (t = 8 h)	pH${}_{e}$ (t = 48 h)
5.00	5.60	6.07
6.00	6.60	6.91
6.40	7.12	7.33
6.80	7.53	7.67
7.20	7.88	8.15
7.40	8.20	8.52

**Table 2 biology-12-01221-t002:** Quantitative values corresponding to the results displayed in Figure 10 and calculation of the cell viability gap presented in Figure 11.

F98 Monolayer (2D)	Extracellular pH	% Cell viability with	% Cell viability after	Cell viability
	(pH${}_{e}$)	DMSO (control)	TMZ treatment	gap
	6.91	39.28 ± 2.38	23.80 ± 3.31	15
	7.33	61.90 ± 3.14	39.28 ± 2.09	22
	7.67	75.79 ± 4.81	28.17 ± 0.68	47
	8.15	105.15 ± 2.47	57.53 ± 3.43	48
	8.52	100 ± 2.99	37 ± 2.47	62
U87 Monolayer (2D)	Extracellular pH	% Cell viability with	% Cell viability after	Cell viability
	(pH${}_{e}$)	DMSO (control)	TMZ treatment	gap
	6.91	47.30 ± 2.74	32.93 ± 2.74	14
	7.33	48.30 ± 3.11	23.95 ± 3.11	23
	7.67	72.45 ± 3.73	50.89 ± 3.73	21
	8.15	83.83 ± 2.07	41.91 ± 2.07	41
	8.52	100 ± 1.03	35.92 ± 1.03	64
F98 Spheroid (3D)	Extracellular pH	% Cell viability with	% Cell viability after	Cell viability
	(pH${}_{e}$)	DMSO (control)	TMZ treatment	gap
	6.07	84.44 ± 7.21	68.27 ± 3.58	16
	6.91	123.13 ± 7.89	62.27 ± 4.72	60
	8.52	100 ± 8.64	29.83 ± 6.81	70
U87 Spheroid (3D)	Extracellular pH	% Cell viability with	% Cell viability after	Cell viability
	(pH${}_{e}$)	DMSO (control)	TMZ treatment	gap
	6.07	80.49 ± 6.89	67.99 ± 5.36	12
	6.91	121.73 ± 6.21	78.11 ± 5.29	43
	8.52	100 ± 3.31	35.62 ± 4.84	64

## Data Availability

Not applicable.

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
