# Peer review of "Characterization of the Intracellular Acidity Regulation of Brain Tumor Cells and Consequences for Therapeutic Optimization of Temozolomide"

_biology, 2023, doi:10.3390/biology12091221_

Round 1

Reviewer 1 Report

The manuscript by Tafech et al entitled " Characterization of the intracellular acidity regulation of brain tumor cells and consequences for therapeutic optimization of temozolomide" could be considered for publication after revision.

Please see below the topics that the authors should consider:

- The authors do not present the statistics in the graphs. Are the differences significant among conditions? Please add statistics to the graphs.

- Pag10Line402: The authors state that U87 showed increased mortality compared to F98. Is this difference statistically different? If yes, would it be worth it to plot both cell lines to highlight such differences?

- How do the authors explain the behavior of F98 under basic conditions? The authors propose the same mechanism for F98 and U87 on basic conditions, but U87 behaves differently under basic conditions.

- The authors used 3D models that have been quite used in the last years between In vitro and In vivo studies. However, the TME is a highly complex ecosystem where different cells can be found. The authors use 3D models to evaluate acidity using only tumor cells. To have a better translation of such observations into in vivo settings, the authors should consider the use of 3D spheroids that have other cells present in the TME.

- Figure 9: Please adjust the Y axis as A and B have the same scale.

- The U87 2D and 3D showed huge differences in the pH. How many spheroids were used to evaluate the pH? Although the authors show the SD, there is no information in Figure10, on the number of replicates used.  

- Figure 10: Please adjust the Y axis as A and B have the same scale.

Are the differences among ROI statistically different? In the case of U87, seems that there are no differences among ROIs. Therefore, the possible explanation presented by the authors is not reasonable. “The fact that the U87 peripheral cells lowered, under acidic conditions, their intracellular pH can be explained by the inhibition of this exchanger, the H+ ions are therefore conserved in the peripheral cells. However, unlike peripheral F98 cells, the decrease in pHi under basic conditions allows us to hypothesize that peripheral U87 cells "overexpress" Na+/H+.”

- Are there other studies reported using TMZ on F98 and U87 cell lines? How were they performed? Could be interesting to add this information in the discussion to understand the added value of using TMZ at the most proper pH.

Reviewer 2 Report

The English is mostly fine I just detected and reported in the revisions small comments about the use of a more appropriate scientific form and less colloquial one

Reviewer 3 Report

Dear Authors,

your manuscript is interesting, still some improvements are necessary in my opinion.

First of all, some amendments in the English language are needed, since some sentences are a bit difficult to read. Please find the sentences highlighted in yellow in the attached PDF of your manuscript.

Second, I suggest you to increase the font size of the Figures, to make readability easier. Moreover, please add statistical significance in the barograms. You have also to add a "Statistical analysis" section to the "Methods".

I would also advice you to include the following papers to the References, because I think they can help to enrich the Discussion. 

PMID: 37568758

PMID: 36638953

Dear Authors,

overall the quality of English is ok, still there are some sentences that are a bit difficult to read. Please find the attached PDF of the manuscript with some sentences which I have highlighted in yellow, and that need to be improved, in my opinion.

Round 2

Reviewer 2 Report

I am pleased with the changes made by the authors after the reviewer's suggestions and the fact they lower the tone while describing their finding. I appreciated they followed the guidelines, and since they made the requested changes I accept their explanations about the intention not to complicate the biological system by adding further components to their study, since no discussion on this regard was made, nor literature involvement was considered.

However I would like the author to consider and explicitly declare that the mechanism their proposed in Figure 13 in based on merely assumptions of their own and none of the mechanisms offered it is actually proven in this manuscript. This sentence/declaration should appear in the discussion, in Figure 13 and in the abstract of the manuscript, although they mentioned that it was hypothesized. Yet, it must be clear to the readers that this content is missing in the author's characterization from the very beginning, and the actual mechanism of pH regulation was not proven. Words like "appeared, assumed, considered, hypothesized",  etc. are not acceptable enough. 

I would then reconsider the manuscript acceptable for publication after these minor revisions will be made.
